# Barriers That Keep Vulnerable People as NEETs

Carlos Pesquera Alonso [1,*] , Almudena Iniesta Martínez [2] and Práxedes Muñoz Sánchez [3]

1    International Doctoral School, Catholic University of Murcia, 30107 Guadalupe de Maciascoque, Spain
2    Psychology Department, Catholic University of Murcia, 30107 Guadalupe de Maciascoque, Spain;
     ainiesta@ucam.edu
3    Education Department, Catholic University of Murcia, 30107 Guadalupe de Maciascoque, Spain;
     pmunoz@ucam.edu
*    Correspondence: cpesquera2@alu.ucam.edu

**Abstract:** The rates of young people Not in Education, Employment or Training (NEET) in the Mediterranean European Economic Area (MED EEA) are high. Hence, the European Union and national governments have developed and implemented different policies aimed to tackle the NEET situation. In this article, we try to identify and understand the most relevant barriers that keep vulnerable people as NEETs. We focused on youth as being at the highest risk of becoming NEETs: migrant women aged 25–29. By using semi-structured interviews and focus groups with key stakeholders and NEETs in the Spanish region of Murcia, we collected their views on and experiences with different programs and strategies. We conclude that this group is exposed to additional barriers due to the intersectionality of their characteristics. There are structural and contextual aspects, such as motherhood, a lack of social networks, or not knowing the language, which NEET policies do not address. We highlight the need of (i) improving the adaptiveness of relevant policies by being tailored to specific problems NEETs face, (ii) expanding the focus of NEET policies, and (iii) including the regional perspective in NEET policy design. In our comprehensive approach, we stress that the NEET policy alone cannot solve the NEET problem.

**Keywords:** NEET; Mediterranean European Economic Area; youth guarantee; woman; migrant; public policy

## 1. Introduction

In the Mediterranean European Economic Area (MED EEA), the rate of young people Not in Education, Employment, or Training (NEETs) increased after the hit of the 2008 financial crisis (Boot et al. 2016). Although these levels have started to decrease in the area since the peak of that crisis, the recovery process was still ongoing when the COVID-19 pandemic stopped it (Eurostat 2022). Recent years have seen the implementation of a series of EU programs aiming for the employability of the younger segments of the population. The success of those programs is still debatable: part of this improvement may be a consequence of the policies implemented in the European Union, but their efficiency is questioned due to their limited success (Focacci 2020). This paper aims at understanding the issues of those measures by analyzing the barriers that keep young people as NEETs, with a focus on one of the most vulnerable groups, migrant women aged 25–29.

After reviewing the research on NEET policy, we identified the special impact that it has on the most vulnerable groups. In order to identify the most relevant hurdles and understand how they are affecting the NEET situation, we performed a preliminary study with secondary quantitative data. This introduced us to the general context, and helped us to identify the characteristics of the most vulnerable people within the NEET category. We later applied a qualitative approach by interviewing both stakeholders who were working on the NEET policy, and NEETs themselves. In this article, we present our findings, and propose including perspectives that also consider other policies and add more adaptability

to the existing NEET policies. Our goals are to contribute to the academic knowledge of the topic, as well as to help the development and improvement of current NEET policies.

Most of the research on NEETs is focused on the national level. However, we argue that the regional level should be considered. Regarding the NEET situation in the MED EEA, Pesquera Alonso and Strand (2020) show that there are substantial similarities between regions from different countries that share similar characteristics, despite the existing discrepancies at the national level. Cefalo et al. (2020) also argue in favor of a place-sensitive approach regarding NEET policies. Thus, this article is one of the first covering the MED EEA from the regional comparative perspective. There are studies which are based on the Mediterranean area, but they usually focus on the national level, and also try to contrast the area with other geographical spaces—sometimes, however, the countries included are too diverse to allow an acceptable comparison between countries and regions (Boot et al. 2016).

Another gap this article tries to cover regards the most vulnerable NEETs. There are studies considering this group (Robson 2008), but they lack a combination of quantitative and qualitative approaches. Some research focuses on specific characteristics (Gökşen and Öker 2017; Maguire 2018), but these studies are narrow to only a small set of characteristics, and none include a whole set of the most relevant barriers that influence the most vulnerable NEETs. In our research, we combine these characteristics to understand how their interactions create hurdles that have a special impact on the most vulnerable groups.

After this introduction, we present a literature review, where we discuss the contributions to the debate and other researchers' findings regarding the barriers that keep vulnerable people as NEETs. It is followed by a presentation of the applied quantitative approach and qualitative methodology for this article. The paper continues with a section covering the most relevant findings. Finally, we discuss the implications of those findings, and propose some recommendations to alleviate the NEET situation for the most vulnerable people.

## 2. Theoretical Background

Although the term NEET is relatively new, there are many policies focused on NEETs (Williamson 2002), and many researchers have already highlighted the importance of these policies to reduce the NEET rate. Following that argument, many countries, as well as the European Union, opted to develop NEET policies to solve the situation (Eurofound 2012b). Although these types of policies may include funds and different mechanisms, as well as strategies focused on reducing the NEET rate, they do not prevent the criticism about resource efficiency regarding their main tools which aim to solve the NEET situation (Rodríguez Soler and Verd 2018). Another highlighted point in this discussion is the range of policies that have an impact on improving the NEET situation. Authors such as Carcillo and Königs (2015) argue that there are policies, which are not NEET policies (e.g., social benefits or special education programs), that are important for the improvement of the NEET situation, especially when one considers the most vulnerable groups. In this context, the authors also remarked that differences in NEET rates are also influenced by different contextual factors, such as the national economic situation.

In the European framework, the main NEET policy is the Youth Guarantee, developed after a 2013 European Council Recommendation. Immediately after its implementation started in the second quarter of 2014, there were authors who claimed that it lacked the level of resources needed to achieve its own recommendations (Escudero and López Mourelo 2015), that to achieve its goals, it should address other factors that were causing the NEET situation (Pastore 2015), and that it could provoke negative side effects (Cabasés Piqué et al. 2016). After several years of implementation, there is no clear conclusion concerning this policy. Many authors present findings that reveal that the policy is partially working (Focacci 2020), but it is still far away from achieving its goals. Others wrote about the adjustments that it experienced (Kraatz 2017), and the time and changes it needed to adapt to each context (Milana and Vatrella 2020).

Research shows that the wider the range of people under a policy, the more difficult it is to adapt the policy to the profiles of the different subgroups (Lahera 2004)—and that more diversified social spending increases the probability of success of people in need (Cuadrado Roura et al. 2017). Hence, many, such as Eurofound (2012a) in its reports on NEETs, support the thesis that the more tailored to the target group a policy is, the higher the probability of its success will be. In order to achieve this ideal of an efficient tailored policy, authors such as Mascherini (2017) highlighted the value of public policy evaluation. The author showed that, regarding the NEET policy, the evaluation process helps to improve the flexibility and adaptability required to successfully achieve the needs of the diverse target group.

In line with the intersectionality indicated by Gökşen and Öker (2017), women show a higher risk of not achieving labor integration when their age increases, or when the focus is solely on migrants. Other authors have remarked that women show greater participation in education, especially during adolescence, but less employability when they enter the labor market (Vancea and Utzet 2018), which is linked to different ideas concerning the role of women in society. Women may have a greater propensity to assume household duties, which translates into a factor that influences the disparity in NEET rates (Maguire 2018). Due to the age group, an important role that influences women and can become a barrier is motherhood. Studies for European countries with different levels of gender equality and NEET rates, such as Finland (Saloniemi et al. 2020) or Hungary (Szabó 2018), included the factor of care status in closing this gender gap. Despite the different approaches and reasons behind the policies existing in each country, the benefits, offered in the form of financial aid or maternity leave, favor that more women than men choose to care for their offspring. As a consequence, some women are left out of the labor and training markets.

Regarding the characteristic of coming from another country, the first barrier that migrants face is related to how the concept of a migrant is socially understood (Brahic and Lallement 2020). Due to this social idea, being labelled as such may bear negative implications, even for those who were born in the country, but descend from people who were not. Being a migrant has real measurable effects—for example, on average, a lower educational level is reached (Borgna 2016). They may face extra barriers, such as not being fluent speakers of the local language or experiencing difficulties obtaining official recognition for their educational certifications, which results in 'disqualification' (Gökşen and Öker 2017). These are elements of vulnerability that, as highlighted by Gökşen and Öker, are reflected in the higher NEET rate for migrants. Another factor that affects them is the community in which they live. Living concentrated in residential areas or so-called ghettos can facilitate comfort in the case of language difficulties, by surrounding themselves with people who speak the same language. However, it is also linked to issues in educational and work integration. Checa Olmos et al. (2018) pointed out that certain dynamics of migrant labor integration caused these difficulties, and they can be transmitted over generations. These authors focused on the primary sector, since it is a space with a high relative percentage of migrant population. They showed that these levels of participation in the sector can be explained by the reduction of barriers such as language, unskilled manual work, the high irregularity of contracts, the physical burden of work, etc.

There is also research signaling barriers that are not exclusive to NEET policies, and do not only affect the most vulnerable people. For example, the motivation of both users and the key informants that implement public policy, especially if the demotivation of public workers reaches high levels, is a relevant factor (de Simone et al. 2018). In the case of NEETs and more vulnerable groups, this motivational aspect may be related to low expectations. They can derive from a lack of confidence in their future, thinking that the economy is performing in a worse way, or that the solutions provided by the institutions are not useful, etc. (Cabasés Piqué et al. 2017). In particular, the migrant population may carry extra considerations (i.e., not knowing the institutions, being afraid of incompatibilities with other aid, or being in an irregular situation), as well as the social pressure and expectations regarding being older or a woman (Maguire 2018). Other difficulties may be related to the

dissemination of programs (Cabasés Piqué et al. 2017), which implies that institutions need to inform and convince the target group to participate in the policy.

## 3. Methodology

The term NEET refers to people with vastly different risks of entering and leaving the category. In particular, the youth at a greater risk are those who share key characteristics that expose them to the worst consequences (Robson 2008). As we described in the previous section, the academic literature identifies one subgroup as especially vulnerable: migrant women aged 25–29 (OECD 2016; Pesquera Alonso and Strand 2020). Therefore, we checked first that it was also a fact in the MED EEA countries. Figure 1 corroborates that the characteristics of migrant women aged 25–29 constituted risk factors in the countries of the MED EEA during the research period. It shows that those who shared these characteristics had a higher risk of being a NEET than a random young person. Before the 2008 financial crisis began, the likelihood of being a NEET for the people with the selected characteristics in comparison with the rest of the youth was at its peak in the MED EEA countries, with the exception of Spain. Figure 1 shows that the situation changed after the crisis hit. However, it bounced after the worst part of the crisis, and this trend continued until the end of the study period.

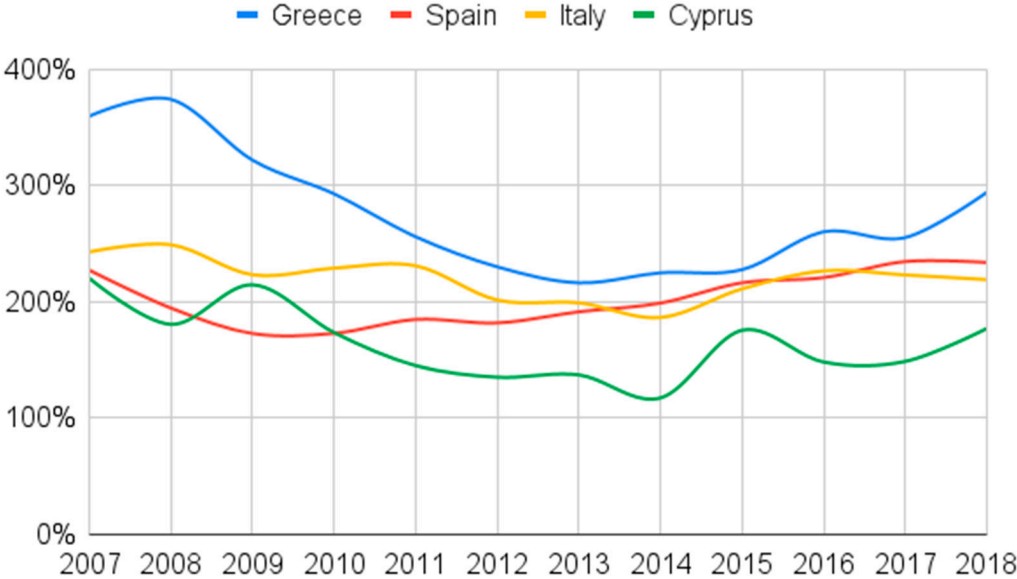

**Figure 1.** Likelihood of being a NEET.

Secondly, in order to better understand the barriers that keep vulnerable people as NEETs, we analyzed the perceptions of key stakeholders whose work was related to this topic, as well as the perspectives of NEETs. To achieve this, we limited the research to the case study of the region of Murcia as a representative region of the MED EEA countries. We prioritized including the regional perspective. Thus, we chose the region of Murcia as a Mediterranean region which has experienced flows of migration, and presents a high NEET rate (Eurostat 2022). We also consider that its economic structure and development presented the region of Murcia as a good representative for the general context of the study countries.

We verified that, due to the strong delimitation of the target group (migrant female NEETs aged 25–29), the target group represents a minority among minorities. In the first quarter of 2022, 85,642 people fell within the description in Spain, 4175 of whom were in the region of Murcia (INE 2022a). Due to the reduced size of the population, there were no stakeholders whose work focused only on migrant female NEETs aged 25–29. Another challenge was that the characteristics of the target group usually imply linguistic barriers, cultural differences, and economic limitations. This entails access and communication

difficulties, both for the investigation itself and to find stakeholders from whom we sought to collect information. Thus, we selected a snowball sampling, due to the characteristics with which we dealt (McKenzie and Mistiaen 2007).

Under these restrictions, for the stakeholders, we established the objective of reaching both those who were responsible for choosing the public policies that affect our target group and those who implemented them. While we gathered the contacts, we interacted with unions, business groups, public institutions, social services, non-governmental associations, employment experts, training companies, and other social partners. Regarding the stakeholders, we compiled 26 in-depth interviews in 6 archetypes. Those archetypes were put into the following categories: public employment Agencies (A), public employment policies Coordinators (C), Non-Governmental Organizations focused on assistance to immigrants (NGOs), managers of Professional Organizations (PO), Work Counsellors (WC), and Trainers (T). The interviewees represented different perspectives: some help migrants specifically (e.g., NGOs), while others work with vulnerable people. In some cases, even approaching certain types of key informants may imply losing the characteristic of being a NEET (i.e., by participating in training or work). Regarding the NEETs, we collected the data through two different methods. The first consisted of 44 face-to-face interviews, with questionnaires consisting of closed and open questions. The second method referred to data collection via a focus group.

Regarding the stakeholders, we carried out interviews during the second week of April 2019, from the last week of September of 2019 to the beginning of December 2019, the second week of July 2021, and the first week October 2021. For the NEETs, we carried out the interviews during the second week of April 2019, and the focus groups on January 21st, 2021. While the interviews took place in the field, the focus group was performed online. The duration of the interviews and the focus group varied between 35 and 87 min. The interviews with the stakeholders followed the semi-structured interview methodology (Valles 2007). We had a script with categories and topics (See Appendix A), and the stakeholders could further discuss according to their experience and knowledge. In line with the logic of the snowball methodology, we included, adjusted, or removed some questions after the interviews, following the stakeholders' experience. This allowed us to collect the key informants' opinions, thanks to the added adaptability of the method (Valles 2007). The interviews with the NEETs consisted of two parts. The first part refers to a structured questionnaire that we are not using in this research. The second part consists of a set of open questions (See Appendix B) that follows the same logic as the previously mentioned methodology for the stakeholders. The methodology for the discussion group (Lobe et al. 2020) follows the recommendations for an online format, since it took place under the limitations of the COVID-19 lockdowns (See Appendix C). In the discussion group, six NEETs participated in the video conversation.

Understandably, each NEET can only provide their personal perspective, but the stakeholders find themselves in another situation. Thus, we remark that the stakeholders may present biases caused by their positions; the public employment Agency interviewees had the most institutional role. We interviewed people with primarily technical responsibilities, but even within the anonymous space of an interview, the political component of their position might influence the interviewees. The public employment policy Coordinators have access to a greater diversity of young people. However, the direct interaction with NEETs affects one's perception of them. The NGOs are the closest to the target group, but the moral principles guiding the entities might affect the perspective of the statements provided. Their link with migrants and institutional dependence should be considered. The Professional Associations are limited by the professional space. The Work Counsellors know the difficulties experienced by those who seek integration help first-hand. It is the position goal, and the reason behind the application for the job—that human perspective may bring one closer to the particular case, but risks forgetting the biases. The Trainers, although dealing with people directly, do not know a lot about the personal experiences of the people they train. They hardly had access to an analysis of the overall situation beyond

what they could extrapolate from the characteristics and their changes in the composition of the groups that attend their courses.

We divided the topics included in the interviews into three different categories: (i) inequality structural factors, (ii) social, economic, and institutional context, and (iii) NEET public policy implementation. Some of those topics are common for all categories (concepts of migrants and NEETs; cultural differences and stereotypes), some are included in two categories (homologation and overqualification lacked in the implementation category; tailored policies in the structural category), and the rest belong to just one of the three categories. Thus, regarding inequality social factors, we sorted the responses in the topics of Social structure, Inequalities in access, Family and motherhood, Work–life balance, Social networks, and Language. The topics related to the social, economic, and institutional context were Institutional trust and knowledge, Integration plans, Primary sector, Social values and expectations, Personal motivation, and Mobility. Finally, in relation to the NEET policy implementation, we divided the contributions from the interviewees in the topics of Youth Guarantee, Dissemination and mediators, European funds, Market feedback, Technical problems, and Work motivation.

## 4. Findings

The first topic, included in all the categories mentioned in the previous section, regards the concept of a migrant. We confirmed the impact of being labelled as a migrant in our research.

> *"The impact of being a migrant is huge. There is a lack of help and a lot of difficulties.*
> *You don't receive support and you suffer a lot of discrimination"*. (NEET)

In some cases, descendants of migrants who were born and grew up in the country are labelled as migrants.

> *"There are many [migrants] born in Spain"*. (WC)

This labelling of people is linked to how each society perceives migrants, as well as to expectations regarding work, education, social ties, etc. The ideas connected to what it is to be a migrant could imply a barrier that can hinder the aimed integration for NEETs. Some of these ideas arise from cultural differences. Both the shock that those differences may produce and the negative reaction to them may complicate market integration by limiting the approach of migrants towards the available solutions. Some of these ideas are linked to the migrants' understanding and expectations regarding work and education.

> *"There are not many migrants on the Youth Guarantee list. They sign up for employment*
> *offices, because they are looking for a job. They do not think about training"*. (A)

In that sense, it is highly relevant to understand the usual situation of many of the most vulnerable NEETs, because, due to their situation, they do not see that they have many options available.

> *"We have to do anything to survive"*. (NEET)

In other cases, those ideas are linked to other cultural aspects, such as religion or traditions.

> *"One of the things that closes many doors is the veil [used by Muslim women]"*. (WC)

Sometimes, these ideas become stereotypes, and they may influence job opportunities considerably. This is more evident regarding migration The stereotypes may limit the opportunities NEETs have to find a job.

> *"People prefer to hire people from East Europe [ . . . ] before migrants from other countries.*
> *If there are none, if they have to decide, they'd rather hire a South American. If there*
> *are no South Americans, they prefer to hire a person of colour. And if there is no person*
> *of colour, then they already hire a Moroccan person. If there are no Moroccans [ . . . ]*
> *Algerians, the last are Algerians"*. (T)

*"Thus, I would only be able to get manual job positions. In the job market, I have experienced discrimination. People do not trust you. They think that maybe you are a bad person".* (NEET)

This issue is also visible when the gender perspective is considered. A job may not be offered to the NEETs because the companies do not expect them to be qualified for the position.

*"Because I am a woman, nobody would hire me for some jobs [ . . . ] a woman can work cleaning and caring while men wouldn't be hired. It depends on the kind of job".* (NEET)

Once established, these stereotypes create structural elements that may generate inequalities for those who share the characteristics of our target group. In particular, we found that the interviewees understand that the success of the integration process depends on the social structure, as well as the opportunities it offers. Nevertheless, in some cases, the opportunities our target group finds are not fit to provide real integration.

*"Traditionally, work was the way out of the situation of exclusion. Today, it is not real. Therefore, they, who choose very low-skilled jobs, continue to be affected by this situation".* (NGO)

*"It is that if you work and you still have to depend on your parents, that is . . . you have to live with your parents... And only to cover expenses. That is like working when you were 16. I don't think you can consider yourself an independent adult in this situation".* (NEET)

Due to the characteristics of our target group, being able to find work that gives them the option to integrate is especially relevant. However, they do suffer from inequality in access. They have comparative disadvantages for cultural and educational reasons; hence they do not have all the options that a random person from outside the studied category may have.

*"Surely outsiders will always have a disadvantage. That's obvious. If we want to accept it or not. A person who comes from another country, who does not know the language well, does not understand the culture, . . . it is always going to be at a disadvantage".* (NEET)

In more extreme cases, the stakeholders highlighted those who arrived in the country without a basic educational level, which limits their opportunities.

*"Many do not finish school. They have no studies. Basically, what they are asking for now, [ . . . ] one of the requirements for all positions is the school certificate".* (WC)

Furthermore, due to their characteristics and the situation they are embedded in, they suffer inequalities in access because they do not know how or have the tools to join the options available.

*"I think there are a lot of female immigrants who do not attend these programs. Some come on their own initiative, others because they are forced because they receive basic income and the other percentage . . . I do not know. And there is another part that the husband may not let her".* (PO)

*"It is difficult because of the lack of support and resources".* (NEET)

In the case of women, these structural factors may also manifest through ideas of family and motherhood. These ideas may become life goals and responsibilities that disproportionately hinder women's future in the labor market—for example, opting for solutions that are not compatible with having a job.

*"Many of the women we had in Youth Guarantee were young women with dependent children".* (A)

*"[Muslim women] don't take the kids to day care".* (WC)

We have seen that it is an effect that goes hand-in-hand with work–life balance measures, but cultural and religious aspects also influence work–life balance. We have seen

that there are barriers related to the options that are offered to them, because they are not adapted to their social, cultural, and family needs.

*"Yes, [women aged 25–29] come when they are invited to be informed about the courses and the training opportunities. However, after that, because of their schedules, they cannot sign up"*. (A)

*"We take into account the work–life balance needs that they may have, either using public resources, orienting them towards the use of those public resources, or we, if this is not possible, create parallel spaces for it"*. (NGO)

Thus, the inclusion of work–life balance measures in any NEET policy is the key in allowing them success. In other words, this policy should address life plans if they want to achieve the goals regarding vulnerable people. It is especially relevant regarding migrants, because some do not consider that they have finished their migration journey, although they may be forced to stay in the country where they first entered.

*"For many, Spain is a stop on a path, not where they want to stay"*. (WC)

*"I would have no problem emigrating to another place. In fact, Germany seems like a good place to go"*. (NEET)

Here, many come to value social networks and support from trusted people. Without social networks, people do not have the basic tools needed to advance within the framework of any NEET policy. Many use those contacts to gain their first jobs, to learn the language, to know how to receive support, etc.

*"I just arrived in this country and the lack of network connections is important"*. (NEET)

*"The final destination for most of the migrants is not Spain. Their goal is mainly in France, in countries of central Europe. [ . . . ] There are their social networks, which are the social networks they need for support and coverage"*. (NGO)

In fact, learning the language is generally the first step towards achieving labor integration. Without the language, it is virtually impossible to reach integration. We have seen that it is sometimes difficult to join these courses, especially for women who have children.

*"First learning Spanish. Without learning Spanish, there is no possibility of joining the labour market"*. (NGO)

*"Local people will always have preference. Always. Especially in unskilled jobs. People who know the language go after"*. (NEET)

*"There are no flexible schedules for these women [with kids] to go to learn Spanish"*. (WC)

In some cases, in addition to the language difficulties, our target group does not receive recognition for the education they have attained. The host country may not accept their documents, and the homologation process may last years or cannot be achieved. This barrier generates cases of overqualification and a waste of human resources. Guided by the need to gain any salary, in many cases migrants with higher education find themselves performing low-skilled labor.

*"I can talk about cases of a Venezuelan lawyer, who has to join high school here right now"*. (NGO)

*"I have met people from Africa and South America. From South America, usually women, who have higher degrees and are training as vegetable handlers and maybe they studied law"*. (PO)

*"Nobody wants you because you have no papers, so you just have to wait. For me it is like starting from zero again. It is as if all my previous studies and training are gone"*. (NEET)

In fact, education policies are a key factor for achieving integration. An unfavorable access to education puts the target group at a greater risk of being a NEET, and hence it is a

barrier that our target group faces. Nonetheless, the NEETs do not always agree on their views on education. Some want to believe in it, and some think that education is not going to provide them better solutions for their situation.

*"My experience has been satisfactory. I found that this kind of education is very useful because it helps to find a job in the sector you want to have a position in"*. (NEET)

*"People, when they finish studying, are not ready for the market"*. (NEET)

Meanwhile, the key informants consider that the vulnerable people are in a disadvantaged position regarding education. They think that the problem should be addressed beforehand, since the NEET policies try to solve it when the damage is already done.

*"Public schools should not be the only ones who accept immigrant children"*. (NGO)

*"Basic training, what the basic educational system should provide, is key"*. (WC)

This shows that in order for vulnerable people to achieve integration, finding a job is not enough. Thus, stakeholders highlight the importance of integration plans for policy success. Those plans are a compilation of different types of policies. However, NEET policies may not consider their interaction with other policies. This also concerns policies such as urban development or neighborhood policies. Regarding this matter, the interviewees highlighted how important those policies are. When asked, they focused on the topics related to other policies, even if they were not directly related to NEET policies.

*"We cannot put all the groups that we label as vulnerable in the same place [neighbourhood]"*. (NGO)

This lack of coordination between policies is counterproductive in generating institutional trust, and it is a kind of trust that those in our target group tend to lack. Some may believe that by joining a program, they may lose benefits or their status in the country. Many may even fear being deported from the country, while others simply do not think that institutions are going to help them to achieve anything.

*"[ . . . ] She comes to the [YG's] first session, you explain to her what it is for and that she comes and tells you that she is not interested at all if you call her, because she works very well with web pages and social networks and others things for job searching, and that this [YG] is a waste of time"*. (C)

*"I have always worked finding work on my own and not through agencies, they have always wasted my time"*. (NEET)

Even if they trust the institutions, sometimes the problem is how the administration works and its complexity. Migrants especially may lack institutional knowledge. Some do not know how to join programs or receive aid, or they think they do not have the right to receive some help while they are receiving support from a different program.

*"One of the main problems of Youth Guarantee is the recruitment of young people, especially young people who are away from any institution, are the most difficult to reach"*. (A)

*"Often there is a lack of knowledge about the existing resources"*. (C)

All of the aforementioned difficulties, along with the needs that they have, have pushed many into the first sector. However, it is not an advantage, but a disadvantage, for our target group. The stakeholders have made clear that opting for that path implies hard physical work and precarious conditions. It seems to be an acceptable option only for those who have no other option.

*"People who get into agriculture stay in agriculture because there is no way out. [ . . . ] Everybody [in agriculture] has been working without a contract"*. (T)

*"Someone tells me that there is something for a day or a week and I do it. It does not matter what it is: gardener, walking dogs, construction . . . They are mainly physical jobs. Those are the options that I have as an illegal migrant"*. (NEET)

Thus, the primary sector does not facilitate integration, nor does it attract young people. NEETs only see those kinds of jobs as badly paid and hard options.

*"The young people themselves have no interest in agriculture either".* (T)

*"[You are only going to be accepted] if you are coming to work in the fields for thirty euros a day. Then perfect, with open arms".* (NEET)

This view of the primary sector coincides with the social values and expectations people have regarding job opportunities in that part of the economy. The stakeholders advised that, in order to alleviate the NEET problem, one should focus on those social values and expectations, especially on elements that are not directly related to the NEET situation, but do have a clear impact on it.

*"I believe it is very important to work with them [migrant NEET women]: education in values, breaking down stigmas, recognizing the worth of women, all those things".* (WC)

To overcome the barriers created by those values and expectations, we must consider the personal motivation that drives the target group members. In some cases, the individual component is clear, and is defined by the interest people have. When someone is really motivated, it can be an immensely helpful way to achieve success within NEET programs.

*"This kind of scheme provides you an opportunity to increase your knowledge. It allowed me to choose things that better fit my profile, more specific things which were more coherent with my motivation and interests".* (NEET)

*"They have a lot of interest and they try hard. People who woke up at 5 AM stayed active in my courses until 10 PM. You see that they are interested in the courses and in training".* (PO)

However, when the individual cases are grouped together, patterns emerge, and one can see the relevance of the policies, context, social structure, etc. For example, the improvement of the general economic situation increased people's motivation. Many started to believe that, after a period without opportunities, they became more positive about finding a job.

*"Now there is a change in motivation and we also see that in the data".* (A)

*"I say, on a theoretical level I have high expectations. The world is beautiful".* (NEET)

However, the effect that COVID-19 had in this trend was considerable, mainly reducing the improved expectations that NEETs have just prior the start of the pandemic.

*"Every day I wake up waiting for the news to say something, but there was never anything".* (NEET)

*"I think the situation is not going to change much. I am positive, but I don't know".* (NEET)

According to the key informants, this personal motivation helps the programs to succeed and increase the opportunities of the people who join the schemes.

*"However, with those who commit and come, in fact, you can move forward with them".* (C)

Another element that has similarities with motivation is mobility. Mobility depends on the personal circumstances, but there are social patterns—for example, the social expectations to find a job nearby, or the willingness to move away to work. The training and job opportunities that NEET policies may offer can be provided in different places, and many young people are reluctant to leave their surroundings.

*"There is still some difficulty in leaving the closest environment".* (A)

*"Of course, they do not consider going outside the town. Don't forget that I'm talking about places like Molina del Segura, which is 5 kilometres away".* (C)

The previous quotes refer to NEETs in general. Nevertheless, the characteristics of our target group are an advantage regarding mobility. In general, it appears that they are less reluctant to move, since migrants usually overcome this difficulty to a greater extent due to their previous experiences moving from one place to another.

*"Because they already emigrate, they have that mobility predisposition and do not have ties to specific people"*. (T)

*"As a migrant, I wouldn't have the problem of moving again, even if I have to move to another country"*. (NEET)

As we mentioned, tailored policies add flexibility to NEET policies. They increase efficiency by providing more adequate solutions to vulnerable people when it is better for them. By adjusting the provided solutions to the individual needs, the policy approaches the real needs of the target group. Thus, the probability to provide an optimal solution is higher. In the interviews, the stakeholders recognized the importance of this approach to achieve better results.

*"To identify what are the capacities, potentialities and, above all, the weaknesses that one has in his/her personal and training environment"*. (NGO)

*"Sometimes there are external circumstances [ . . . ] You are working with a person who has other problems. Sometimes it is not their time"*. (WC)

In comparison, the NEETs have a more personal approach, as their individual experience does not provide them with a more holistic perspective. Even so, they are also aware of the importance of tailored policies.

*"[Talking about experiences with tailored policies] Yes, it really is very practical. Because it teaches you to create a business. Not a company. To open a store. Something that can really feed you. If you do it right, you put in hours and such... You earn a salary and more"*. (NEET)

Providing more personalized solutions is something that the Youth Guarantee aims to do. Due to the novelty of the policy, it initially had to deal with important difficulties, which did not help the NEETs. Its context changed, and so too did the policy which needed to be adapted based on the first field experience. Thus, it entered an adaptation process in the countries and regions where it was implemented. The key informants recalled this experience, and remarked that it is still an ongoing process.

*"When it started there were many problems, the registration issue was especially problematic. This discouraged many young people from approaching the Youth Guarantee"*. (A)

*"However, after 5 years, there is still the negative connotation that it is a program for people with practically no qualifications. People with qualifications do not easily attend"*. (C)

One of the difficulties the Youth Guarantee, as with any NEET policy, faces is dissemination. Not knowing the existence of a policy that aims to help is a clear barrier. Therefore, the interviewees considered it to be a key element. Time is needed to spread the word. Many strategies are used to achieve dissemination, but, sometimes, that information does not reach the target audience.

*"Then you have to go out with a bow to catch them. It is not a matter of the first year, we are observing this for three years"*. (C)

*"There are many people who do not know that these resources exist. They are not promoting it as much as they should"*. (WC)

One element that may help to solve dissemination issues is the use of mediators. The stakeholders agreed that mediators and personal interaction are particularly important in carrying out a more effective dissemination. These are key figures connected to the target group, and mediators can inform and convince our target group to make use of these resources.

*"They do not know the available resources and that is also very important. That's the reason why they need to repeat it again, for mediators"*. (WC)

*"The most effective is word of mouth"*. (NGO)

From the perspective of the NEETs, the mediators' function is not only dissemination, but also to certify the credibility of the policy. This is because the real issue is the aforementioned problem of lack of institutional trust.

*"But those things [offers from institutions] . . . they seem like garbage opportunities to me"*. (NEET)

*"They [employment agencies] don't help you"*. (NEET)

Another relevant aspect of the implementation of NEET policies is how they are funded, as well as whether those funds are used correctly. As previously-mentioned authors have remarked, the stakeholders criticized the misuse of some European funds. One element that the interviewees highlighted in particular was the negative effect of rewarding the participation in some training courses. This option may work as a short-term solution, but it becomes a wasted resource in the long-term, and, in some cases, it does not even work to attract people to the programs.

*"If a person in need is going to earn €900 a month in a project that lasts a year, maybe they train even if they don't like it"*. (A)

*"It happens. We have courses that, although trainees get paid, nobody participates"*. (C)

In order to avoid these types of errors, institutions have tools to help them to adapt the programs. A common tool is the inclusion of market feedback. This tool aims to help them know the needs of the market. Thus, the businesses' role in searching solutions for the NEET situation is usually included. It is usually considered a win-win strategy, since it helps the market needs, includes labor and training offers that may result in labor integration, and it is perceived as useful by the recipients.

*"Every year we collect information from entrepreneurs from relevant sectors to see where the training trends are going. With that information, we organise the courses"*. (A)

*"It's good, because there you have something like a job, which is like between work and an internship. But it is paid. Then it gives you the possibility to live in the country where you are going to do that job. And it is also a job that is related to your interest and what you want to do in your life"*. (NEET)

There are other elements that aim to improve the policy and decrease the barriers that may constitute problems for the NEET policies. The experience with the policy is one of them, since there can be technical problems. During the interviews, the stakeholders signaled some of the problems, which implicitly revealed the importance of adding adaptability to the policies.

*"We need a tool [common national database]. We work in a globalised world, we have e-government . . . It is sad that we use sticks and stones"*. (C)

The last element that can become a barrier is the motivation of those implementing the policies. The output of the policy does not simply become a number for them: it also affects the key informants. In some cases, depending on how the process takes form, it produces a loss of motivation. We found that the stakeholders really want the policy to succeed, and for the NEET rate to decrease. However, the results they see and what they experience do not always contribute to maintaining that motivation.

*"Honestly, after so many years of fighting so much, I don't see the effort it takes, I don't see the results it should bring"*. (C)

*"If these things [NEET policies] don't change, the situation will continue to be the same"*. (WC)

## 5. Discussion and Conclusions

In the analysis above, we discussed some of the most relevant barriers keeping vulnerable people as NEETs. As such, the stakeholders pointed out that, when establishing a public policy that tackles the NEET problem (even more if we take our target group of migrant NEET women aged 25–29 into account), one should not focus solely upon the

training and working options, but one should also consider structural issues and the social and economic context, in line with the remarks of O'Reilly et al. (2018). This perspective highlights that the target group is complex and affected by many diverse aspects, as the NEETs themselves confirmed.

We identified overcoming the linguistic barrier as the most basic priority to start the integration process (García Juan 2015). Both key informants and NEETs agreed that this difficulty is the one that limits the employment options of the group members the most. However, those who arrive in the host country speaking the same tongue do not suffer from this problem. For example, most of the migrants from Latin America who arrived in Murcia were already fluent in Spanish, the local language.

We also identified work–life balance as a relevant factor. One could link the target group with the idea of mother and caregiver, especially considering that we focus on the age group of 25–29 years, when migrant women present high pregnancy rates (INE 2022b). Thus, the availability of the target group is disrupted when the training and job offers are not compatible with their schedules. Regarding that topic, in accordance with the research and recommendations of Molloy and Potter (2015), it is essential to include flexibility for their work–life balance. However, this work–life balance problem is not unique to motherhood and care. The schedules are usually not adapted to cultural and religious customs of minorities rooted in some members of our target group. For example, for female Muslim migrants, respecting the month of Ramadan may not be possible. This shows the importance of considering the NEET's identities (Pesquera Alonso et al. 2022), and this type of issue makes it difficult to combine identity with labor market integration.

We presented research showing that some social policies generate situations that are difficult to alleviate afterwards with training policies. In particular, the stakeholders agree that a housing policy that prevents the consolidation of so-called ghettos (Sim et al. 2003) will help to avoid creating spaces where following generations reproduce the same patterns of poverty. This connection between policies ratifies what we, as in Valtonen (2016), showed: achieving labor integration largely depends on previous social integration. Therefore, public policies that facilitate social integration, allowing interactions between people from different strata, are essential, so that subsequent generations do not remain in the same situation as the migrant population. These elements have an effect on fulfilling the basic needs of the NEETs, and this research shows that fulfilling basic needs motivates NEETs more than the desire for personal self-realization. Thus, public policies that act on social aspects can create a space which is more likely to boost the desire to end being a NEET.

Regarding the issue of mediation, it is an essential factor, both to know the needs of the target group, and to publicize programs (della Noce et al. 2002). In many cases, dissemination is performed through word of mouth. According to the information we collected, the entities that work with migrants, women, and/or young people spread the message better than official institutions, but worse than community figures. This group of NEETs scarcely trust established institutions, and these entities and community figures help to avoid that problem, and are useful when translating ideas, from the general objectives of the programs to the specific needs of the people. This perspective is also shared by Simões et al. (2022), whose research highlights the importance of setting emotional bonds with the community as a way to improve the inclusion of NEETs.

Thus, the findings revealed that the policy should move from having a general perspective of the NEETs and their situation to providing more flexible solutions. We showed that the actual framework for NEET policies in the European Union, the Youth Guarantee, adds some flexibility when it is properly combined with the market. However, not all authors support this idea. For example, (Cabasés Piqué et al. 2016, 2017), see this process from a more pessimistic perspective: "Demand-side policies should then stop to promote precarious jobs, focusing instead on the creation of quality employment" (Cabasés Piqué et al. 2016, p. 699), although it produces key synergies with companies. In this case, the remark of (Cabasés Piqué et al. 2016, 2017) should not be forgotten, because, due to the

economic vulnerability of this group, its members priorities work integration rather than motivation or interest.

The primary sector may seem to be a suitable option to provide jobs for our target group, since there is always demand for workers. In that sense, the work of Baselice et al. (2021) confirms that assertion. They highlighted the opportunities that this sector may offer to NEETs is important. Another factor that has an impact in this option is mobility, which is linked to that type of work, because the sector is strongly connected with jobs outside of cities. Thus, in the case of migrants, due to their roots, they are usually more likely to overcome this difficulty. However, the demand for workers is constant, since work in agriculture is considered to be harder and less remunerated than in other parts of the economy, making the sector less attractive. The hardness and difficulties mentioned here are evident for both stakeholders and NEETs, as Simões also noted in his research on NEETs (Simões 2018). In other words, the entrance barriers are low, especially for migrants, because the precariousness is high. Thus, although it may serve to reduce the number of people in the target group, under the current conditions, it is probably not an integration opportunity. We consider it important to remark that the main policy in that sector, the Common Agriculture Policy aims to rejuvenate the countryside, mitigate climate change, etc., and it depends on the EU level, which provides community funds. Therefore, we consider it essential to link the primary sector with direct employment, but also with rural development, especially because there are many successful programs with this focus (Hoffmann and Hoffmann 2018; Simões 2018). It means that these hurdles are surpassed if the policies connect issues, such as maintaining the environment or rural tourism. To achieve this, collaboration with involved stakeholders and reinforcement of the associative fabric is as important as direct labor integration. Despite the mentioned difficulties, the positive aspect is that it is a space ready to receive this group.

The findings of our research, as well as the relevant literature, point to the fact that there are many barriers that keep vulnerable people as NEETs that do not fall under the scope of NEET policies (Molloy and Potter 2015; O'Reilly et al. 2018; Focacci 2020). Thus, to achieve the goals of those policies, they should be combined with other policies that include social measures, language training, the collaboration of social companies, and cultural integration, etc. If not, there is a risk of not meeting the desired objectives at the cost of continuing to waste talent, work, and public financial resources. Furthermore, the policies should be tailored to add the needed adaptability to achieve their goals. This means that the policies should be nearer to the individual, which goes hand-in-hand with including the regional and local perspective to the policies. However, as we mentioned above, the interviews revealed many structural factors. We divided them into the topics of Concept of migrants, Cultural differences and stereotypes, Homologation and overqualification, Social structure, Inequalities in access, Family and motherhood, Work–life balance, Social networks, and Language. They condense some of the most important barriers our target group faces, and can be combined with technical problems regarding the implementation of the NEET policies. Nonetheless, these policies aim to tackle only a narrow issue in comparison with the general economic and social framework. Thus, as the stakeholders realized and the NEETs confirmed, NEET policies can alleviate the situation, but cannot solve the problem if the structural factors are not addressed.

**Author Contributions:** Conceptualization, C.P.A.; methodology, C.P.A. and P.M.S.; validation, C.P.A., A.I.M. and P.M.S.; formal analysis, C.P.A.; investigation, C.P.A. and P.M.S.; data curation, C.P.A.; writing—original draft preparation, C.P.A., A.I.M. and P.M.S.; writing—review and editing, C.P.A., A.I.M. and P.M.S.; supervision, A.I.M. and P.M.S.; project administration, C.P.A.; funding acquisition, A.I.M. and P.M.S. All authors have read and agreed to the published version of the manuscript.

**Funding:** This research was funded by EEA and Norway Grants Fund for Youth Employment under Grant 2017-1-345.

**Institutional Review Board Statement:** The study did not require ethical approval.

**Informed Consent Statement:** Informed consent was obtained from all subjects involved in the study.

**Data Availability Statement:** The data presented in this study are available on request from the corresponding author. The data are not publicly available due to privacy concerns.

**Conflicts of Interest:** The authors declare no conflict of interest. The funders had no role in the design of the study; in the collection, analyses, or interpretation of data; in the writing of the manuscript, or in the decision to publish the results.

**Appendix A**

Script of the questions for the stakeholders divided by topics:

- The importance of the social economy in the integration of NEETs;
    - Does this business contribute to training and offering jobs?
    - How do they provide feedback to the public institutions and trainers?
    - Is it perceived as a relevant factor?
    - Knowing that it is very important in the first sector, how well do they integrate NEETs?

- The role of the primary sector as a training and labour integration space;
    - Is it a sector that asks for jobs?
    - How is it socially considered from the NEETs' perspective?
    - What kind of conditions are offered?
    - What is the importance of the rural essence of the 1st sector in NEETs integration?

- The effects and importance of European funding on the programs aimed at young, unemployed and migrant people;
    - Have they increased?
    - Do the EU funding decide which programs are implemented or do they depend more from the national/regional/local level?
    - Is it efficiently distributed and implemented?
    - Is it enough?

- On the types of entry profiles (more common and more and less successful);
    - Do people come from different sectors?
    - What are the kind of successful jobs people get into?
    - Are there people from one sector that directly do not apply into training programs?

- On the perceived success of NEET public policy;
    - Do people trust the training programs?
    - Do NEETs enter some programs because of direct profits (paid training programs) instead of looking for the training itself?
    - Does the Administration implement this policy believing it is going to be successful or does it look for the short term money input?

- On the participants' motivation before, during and after participating in the various programs;
    - Do NEETs believe they are going to get a new job after the training?
    - Do they enter the programs without nothing if it is worthy?
    - Do they leave the programs before they finish?

- On the changes on the former educational/labour paths of the trainees;
    - Do they decide what labour path to follow or is it decided by the programs or the economy?
    - Are people reluctant to move from one sector to another, to enter a training program of a different sector?
    - Do they successfully reorient their paths?
    - What are the different perceptions of the NEETs depending on their educational background?

- On the effective feedback mechanisms included in the programs and how they work;
  - ○ Where does the feedback come from?
  - ○ Is it considered into the policy?
  - ○ Is the policy flexible enough to take it into account?
- On the approaches (bottom-up or a top-down) of the implementation follow related to NEET programs;
  - ○ Does the design come from a more governamental perspective or from the NEETs themselves?
  - ○ Do the different sectors have a say into the policy?
  - ○ how have the training programs been modified over time and geographically in the region;
  - ○ Do the training programs adapt to the economical changes?
  - ○ Are they too stable compared to the changes in technology?
  - ○ Do they follow a common national education and training plan or a more regional/local one?
- On the incorporation of direct economic aid in the programs;
  - ○ Does direct economic aid destroy the motivation of the people?
  - ○ Does the topic become irrelevant when people only join because of economic aid?
- On the dissemination of the information of the different programs;
  - ○ Are there different spread methods depending on gender and nationality?
  - ○ How important are websites compared to mouth to ear?
  - ○ Do businesses help to spread the word?
- On the stratification of the programs and on the profiles of the applicants;
  - ○ Is there diversity in the programs?
  - ○ Do the NEETs divide themselves depending on the kind of training?
  - ○ Do migrants enter into different kinds of training than nationals?
- On the interaction between institutions, companies, associations, trainers and individuals;
  - ○ how responsible/independent are the companies/state in terms of the training/employment offers;
  - ○ Can companies influence the administration to offer special training courses?
  - ○ Do the companies invest in training or do they wait for the government to do that?
  - ○ Can the national government decide or does the EU decide?
- On differences between immigrant and native profiles;
  - ○ Do nationals and migrants have different backgrounds and how does it influence their probabilities to be NEETs and integrate into the labour market?
  - ○ Are nationals ready to work in the 1st sector?
  - ○ How geographical mobility helps migrants to move to more rural areas?

**Appendix B**

Open questions included in the questionnaire for the NEETs:

- Please describe briefly your experiences in the course of your education.
- Please describe briefly your experiences from training schemes, beyond formal education, you may have participated.
- Why are you currently not employed?
- If you are not seeking employment, education or training, what is the reason? Please describe briefly the situation.
- If you are seeking employment, what is making it hard for you to find a job? Please describe the difficulties/prospects in finding a job in your local labour market.

- When you were younger, what did you want to do upon growing up? Please describe your aspirations and dreams.
- So now you are older. Do you still have the same aspirations/goals? Are you planning to seek/find a job in another local labour market than the one you currently live in? Please define in which local labour market (of your country of residence or abroad) you wish to migrate.
- What do you think is missing to encourage you to take further training/school and/or seek employment? (focus on state employment policies).
- What is the impact of being a woman on your educational and working life? Please describe briefly your experiences and specific real situations/incidents.
- What is the impact of being a migrant/refugee on your educational and working life? Please describe briefly your experiences and specific real situations/incidents.
- Please describe briefly the ways in which you are making a living.
- Please describe briefly the opportunities and obstacles that youth in your city/locality is faced with when trying to gain access into the labour market. Do you see any difference or resemblance to the capital city? (If the participant lives in the capital city, then the NEET should speak about the opportunities and obstacles of the capital in relation to the periphery).

**Appendix C**

Script of the questions for the focus group with NEETs divided into topics

- Individual/social responsibility:
  - In general, why do you think there are so many young people who neither study nor work? Is it more because of the people or is it because of the general situation? Why?
  - Do you identify as NEETs? Why?
  - How do you think society perceives you (failed, lazy, opportunistic, unlucky, ...)? Why?
  - In your opinion, which are the reasons for being in this situation of not having a job or being in training?
  - Do you think it's your fault? Why?
  - Do you think that the economic situation is to blame? Why?
  - Have you previously looked for work and found nothing? Why do you think it was unsuccessful? How long have you been in that situation?
  - Under which conditions would you accept a job (the level of salary, close to home, the type of work, etc.)? Do you think you ask for too much? Do you think that there are hardly any jobs with those conditions? Why?
  - If your situation were different (you had more or less resources, your families could not support you, you were more or less young, etc.), do you think you would be in the same situation? Why?
  - If your social/economic situation were different (the economy was as before the crisis), do you think you would be in the same situation? Why?
- Expectations:
  - How has COVID-19 affected your expectations? Why?
  - Do you think you will find a job soon? Why?
  - Do you think you will find stable work? Why?
  - Do you think that the general situation of the economy will improve, worsen or be more or less the same? Why?
  - Do you think that your particular situation will improve, worsen or be more or less the same? Why?
- Policies:
  - Do you think that programs aimed at people in your situation work? Why?

○      Have you participated in a similar program before? If so, was it helpful? Do you see this project differently?

○      Why did you join this project? Has it met your expectations? How?

○      Do you think that these programs favour social inclusion?

○      Do you think it will be useful to you? Why?

○      Do you trust the employment agencies to find a job? Why?

○      Do you think these programs should focus on gender? And as for where do people come from (whether they are migrants or not)? And should they consider the personal situation of each person?

○      What do you think should be done to improve your situation?

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
