# Peer review of "Barriers That Keep Vulnerable People as NEETs"

_socsci, doi:10.3390/socsci11060253_

Round 1

Reviewer 1 Report

It is appropriate to appreciate the authors' efforts and their decision to pay attention to just such a topic.

The urgency of the topic emphasizes the need to carry out truly responsible scientific research. The authors conducted interviews with selected stakeholders. However, although these are certainly professionals in their field, the article examines only the subjective views of a small group of experts (5 respondents). For the real scientific weight of the whole study, it would be required to examine the views and real motives of the young people themselves as characteristic of NEET. In the current version of the study, it only conveys the alleged views, reasons and motives of NEETs. However, it does not acquire and examine their real statements and opinions.

Therefore, I highly recommend to include NEETs in the study and then compare their opinions with the opinions of the interviewed professionals. Although it will certainly be difficult to obtain such a sample of NEETs, it is essential to achieve the desired scientific quality of the study.

Good luck!

Author Response

Dear Reviewer,

First of all, thank you for your review. It has been very useful for improving the article.

We have compiled a detailed response in the attached .pdf. Please see the attachment.

Yours faithfully,

The authors

Reviewer 2 Report

The objective of this paper is to identify and understand the most relevant barriers that keep vulnerable people as Not in Education, Employment or Training (NEETs). This is an interesting topic, and the paper is fairly well written, though there are a few grammatical and typographical errors.

The paper investigates an important and timely topic. However, the study is based on a very small number of semi-structured interviews. This is a main weakness of the study. Instead of focusing only on the opinions of five professionals who have been interviewed, it will add value to the paper if the opinions of NEETs are also included.

Though the authors have stated that a preliminary study was conducted using secondary quantitative data, it is limited only to presenting the data in Figure 1. Since Figure 1 covers data for 12 years, additional analysis could have been carried out using the data for four countries included.

The methodology of the paper also not discussed well. Conducting interviews and collecting primary data is appropriate for a study of this nature. One of the main weaknesses of the study is not including the opinions of NEETs. The results of the paper are not presented clearly and analyzed appropriately. The conclusions adequately tie together the other elements of the paper. However, authors have not compared the findings of this study with that of previous studies.

Author Response

(The authors gave the same response as above.)

Round 2

Reviewer 1 Report

Congrats to authors with their improved version of paper. It is really well-done.

Author Response

Dear reviewer,

Thank you very much for your support. We appreciate that your comments help us improve the paper.

Yours faithfully,

The authors.

Reviewer 2 Report

The objective of this paper is to identify and understand the most relevant barriers that keep vulnerable people as Not in Education, Employment or Training (NEETs). The revised paper shows some improvements over the first version of the manuscript.

Authors have addressed all of my concerns about the paper. As pointed out in my previous review, the main weakness of that the study is based on a very small number of semi-structured interviews. In the revised version of the paper, authors have added the opinions of a few NEETs who have been interviewed. It improves the quality of the paper somewhat.

Authors have also discussed the methodology in little more detail in the revised version of the paper. Though conducting interviews and collecting primary data is appropriate for a study of this nature, some type of quantitative analysis could have included in addition to presenting the opinions of stakeholders and a few NEETs. However, without knowing the types of questions asked and the types of data collected, it is difficult to comment on this. If the questionnaires used for the interviews of the two groups are included in the Appendix to the paper, readers will get to know the type of questions included in the survey.

Author Response

Dear reviewer,

Please see the attachment with the response to the review.

Yours faithfully,

The authors
